# SSD: Spatial-Semantic Head Decoupling for Efficient Autoregressive Image Generation

## Abstract

Autoregressive image generation models like Janus-Pro produce high-quality images, but at the significant cost of high memory and ever-growing computational demands due to the large number of visual tokens. While KV cache compression has been extensively studied in language modeling, it still remains largely unexplored for the image generation domain. In this work, we begin by identifying a distinct and prominent attention phenomenon, which we term spatial locality and emergent semantic sink. To leverage this key insight, we introduce a novel KV cache compression framework. Specifically, we compress the KV cache for all visual tokens by adaptively decoupling attention heads into two separate types: for spatial-locality heads, our method maintains a short recent token window; for semantic-sink heads, it strategically preserves a compact set of highly-attended tokens. Our extensive experiments demonstrate that the proposed method achieves a 5× reduction in memory usage and a notable 6.6× speedup in overall throughput with only minimal visual quality loss, thereby enabling highly efficient native autoregressive image generation on resource-constrained hardware.

## 1 Introduction

Recent advancements in autoregressive (AR) multimodal models (Tian et al., 2024; Sun et al., 2024; Chern et al., 2024; Wang et al., 2025; Ma et al., 2025), especially unified AR models (Hurst et al., 2024; Team, 2024; Wang et al., 2024; Chen et al., 2025) have revolutionized high-fidelity image generation from textual descriptions. Models like Janus-Pro (Chen et al., 2025) demonstrate remarkable capabilities in producing photorealistic images by treating image generation as a sequence prediction problem within a unified decoder-only architecture. This approach simplifies the generation pipeline by eliminating the need for separate encoders or decoders, while maintaining strong performance across diverse visual domains (Zhang et al., 2025a; Mu et al., 2025).

However, this remarkable success comes at a significant computational cost (Fan et al., 2024; Xiong et al., 2024). The autoregressive generation of high-resolution images and large batches requires processing an enormous number of visual tokens, which often leads to substantial memory demands and prohibitively prolonged inference times, especially for long sequences (Liu et al., 2024a).

This linear scaling of memory usage with the number of tokens, which grows quadratically with resolution, poses a critical bottleneck and severely limits the practical deployment of these models, especially at large batch sizes. While KV cache compression has been extensively studied for language modeling tasks (Xiao et al., 2023; Zhang et al., 2023; Li et al., 2024), these techniques remain largely underexplored within the visual token generation landscape.

Language-oriented approaches typically exploit the sparse attention patterns and token redundancy found in text, while visual tokens exhibit fundamentally different characteristics—they maintain strong spatial relationships and display structured attention patterns that existing methods fail to capture. Consequently, the visual token KV cache, which constitutes the majority of the total cache during image decoding, remains a primary uncompressed bottleneck.

In this work, we address this limitation by identifying a novel attention phenomenon unique to visual token generation: the coexistence of *spatial locality* and *semantic sink*. Through careful analysis of attention patterns in native visual decoder-only autoregressive models, we observe that certain attention heads specialize in processing local spatial relationships (spatial-locality heads),

while others focus on preserving globally significant semantic information (semantic-sink heads). This structural dichotomy presents a unique opportunity for efficient cache compression.

Leveraging this insight, we introduce a novel KV cache compression framework that dynamically differentiates between spatial-locality and semantic-sink attention heads. For spatial-locality heads, we maintain a compact sliding window of recent tokens to preserve local structural information. For semantic-sink heads, we preserve a minimal set of highly-attended tokens that serve as semantic anchors throughout the generation process. This dual-strategy approach achieves substantial memory reduction without compromising image quality.

Our comprehensive experiments demonstrate that the proposed method reduces KV cache memory usage by $5\times$ and improves decoding throughput by $6.6\times$ at the batch size of 128, while maintaining negligible performance degradation on standard image generation benchmarks. These advancements enable practical high-resolution image generation on hardware with limited memory resources. The main contributions of this work are as follows:

- We identify and characterize a novel dual attention phenomenon in autoregressive image generation models, termed *spatial locality* and *semantic sink*, which reveals fundamental differences between visual and linguistic attention patterns.

- We propose a specialized KV cache compression framework that exploits the structural properties of visual attention by applying differentiated compression strategies to spatial-locality and semantic-sink attention heads.

- Extensive evaluation shows our method achieves $5\times$ memory reduction and $6.6\times$ speedup with minimal quality loss, advancing the practicality and accessibility of autoregressive image generation thereby paving the way for its wider adoption.

## 2 RELATED WORK

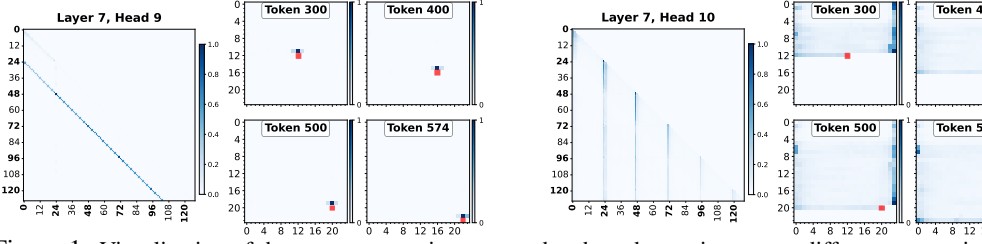

Figure 1: Visualization of the average attention map and reshaped attention across different generation steps in Janus-Pro-1B over 100 prompts, randomly sampled from Geneval (Ghosh et al., 2023). Our observation reveals: (1) Attention in the vision modality is sparse. (2) There are two types of sparse attention: one termed spatial locality attention, the other termed semantic sink attention. See Appendix F for more details.

**Autoregressive Visual Generation and Unified Models** Recent advancements in autoregressive modeling have increasingly extended the proven scaling laws of large language models (LLMs) to diverse visual generation domains (El-Nouby et al., 2024). These approaches typically discretize images into visual tokens via a tokenizer, enabling sequential prediction using decoder-only transformers in a manner similar to text generation, thereby leveraging established architectures.

Pioneering works such as DALL·E (Ramesh et al., 2021), Parti (Yu et al., 2022) and LlamaGen (Sun et al., 2024) have demonstrated increasingly impressive capabilities in high-fidelity image synthesis and instruction following, with SimpleAR (Wang et al., 2025) further demonstrating leveraging post-training refinement through reinforcement learning. A prominent research direction extends this paradigm by unifying multiple modalities within a single autoregressive framework. Models such as Chameleon (Team, 2024), Emu3 (Wang et al., 2024), and Janus-Pro (Chen et al., 2025) adopt a purely decoder-only architecture to process interleaved multimodal sequences (e.g., image-text-text or image-text). Unlike alternative strategies that employ diffusion models, continuous representations, or multi-stage cascaded systems, these increasingly popular native autoregressive unifiers emphasize end-to-end training on discretized token sequences, leading to stronger multimodal synergy and more coherent conditional generation.

Our work aligns with and extends this native autoregressive research line, with a specific focus on improving the inference efficiency image generation—a key challenge for real-world deployment of these unified models.

**KV Cache Compression** Autoregressive decoding in transformer models is inherently memory-bound due to the linear expansion of the key-value (KV) cache during inference. Prior research has explored various strategies, including eviction (Zhang et al., 2023; Li et al., 2024), merging (Zhang et al., 2024; Yao et al., 2025; Li et al., 2025), and quantization (Liu et al., 2024b; Dong et al., 2024) to mitigate this memory footprint.

Early methods, such as window attention (Beltagy et al., 2020), H2O (Zhang et al., 2023), and StreamingLLM (Xiao et al., 2023) primarily focused on decoding-stage cache compression in language models. More recent approaches, including SnapKV (Li et al., 2024), Ada-KV (Feng et al., 2024), PyramidKV (Cai et al., 2024), and DuoAttention (Xiao et al., 2024), have targeted long-context scenarios, emphasizing improvements in prefilling efficiency for large language models (LLMs). Furthermore, while existing methods leverage sparsity patterns inherent in linguistic data, their effectiveness in the visual domain remains underexplored, primarily due to the differing underlying mechanics and distinct token distribution. Other related efforts, such as HACK (Qin et al., 2025) focus on next-scale prediction in visual generation models (Tian et al., 2024), representing a specialized approach to KV cache optimization in hierarchical autoregressive frameworks.

To our knowledge, we present the first novel eviction framework specifically designed to exploit these visual-specific structural properties, enabling efficient and high-fidelity native unified visual autoregressive generation, thereby paving the way for its broader adoption.

## 3 METHODOLOGY

This section presents our methodology for efficient KV cache compression in visual autoregressive generation. We begin by formalizing the problem setup and reviewing fundamental concepts of visual autoregressive image generation with Classifier-Free Guidance (CFG), highlighting the computational bottleneck that motivates our work.

Subsequently, we introduce a key empirical observation: semantic information from textual prompts is preferentially injected into specific spatial regions—particularly the margin columns of the raster-scanned image token sequence. This finding naturally leads to the identification of two distinct types of attention heads with specialized roles: *semantic heads* that capture global context and *spatial heads* that handle local dependencies. Leveraging this structural dichotomy, we then present the SSD framework, which applies asymmetric compression policies tailored to each head type, yielding substantial memory and computational savings while preserving generation quality.

### 3.1 PRELIMINARIES: NATIVE AUTOREGRESSIVE VISUAL GENERATION

Native autoregressive models for visual synthesis, as exemplified by Janus-Pro (Chen et al., 2025), reconceptualize image generation as a sequential token prediction task analogous to text generation. Unlike diffusion-based approaches that employ iterative denoising procedures, AR models generate images token-by-token following a raster scanning order.

Formally, given an input textual prompt tokenized into tokens $\mathbf{T} \in \mathbb{R}^L$, the model autoregressively generates a flattened sequence of visual tokens $\mathbf{Z} = (z_1, z_2, \ldots, z_N)$, where $N = h \times w$ corresponds to the target image resolution in token space (e.g., $N = 576$ for a $24 \times 24$ grid). Each token $z_t$ is predicted conditional on all preceding visual tokens $z_{<t}$ and the text conditioning $\mathbf{T}$. Finally, visual tokens $\mathbf{Z}$ are decoded by a visual decoder (Yu et al., 2021; Esser et al., 2021) to image $\mathbf{X} \in \mathbb{R}^{H \times W \times 3}$, where $p = H/h$ denotes the compression ratio.

A critical enhancement for improving output fidelity and prompt alignment is Classifier-Free Guidance (CFG), which operates through a dual-forward-pass mechanism at each decoding step:

1. Conditional pass: Computes the probability distribution $\mathbf{p}_t^c = p(z_t | \mathbf{T}, z_{<t})$ when conditioned on the text prompt.

2. Unconditional pass: Computes the distribution $\mathbf{p}_t^u = p(z_t | z_{<t})$ without text conditioning.

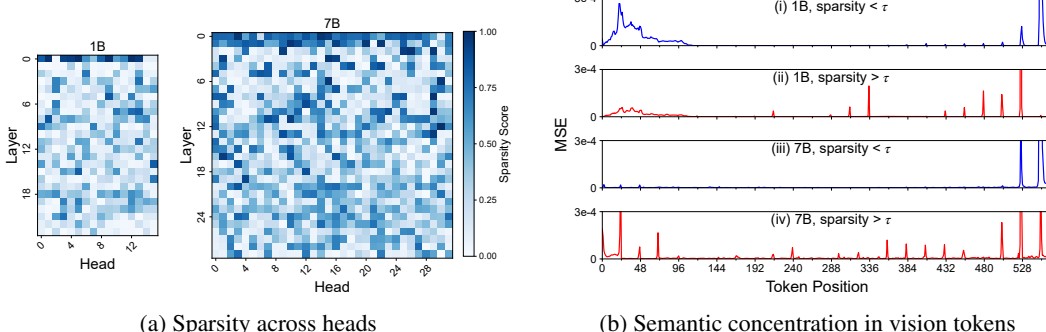

(a) Sparsity across heads      (b) Semantic concentration in vision tokens

Figure 2: (a) Attention sparsity patterns in Janus-Pro aggregated over 100 instances from Geneval (Ghosh et al., 2023) and DPG-Bench (Hu et al., 2024), revealing different attention sparsity across different heads. (b) Token-wise MSE between conditional and CFG-augmented branches, with higher MSE indicating greater semantic information concentration. The periodic spikes in dense heads (red) correspond to margin column positions, confirming semantic anchoring at spatial boundaries.

The guided distribution is obtained by amplifying the difference between these distributions:

$$\mathbf{p}_t^{\mathrm{cfg}} = \mathbf{p}_t^c + \gamma(\mathbf{p}_t^c - \mathbf{p}_t^u), \quad \gamma > 1, \tag{1}$$

where $\gamma$ modulates the guidance strength, effectively enhancing alignment with the textual prompt. The autoregressive decoding process necessitates maintaining a growing Key-Value (KV) cache for all previous tokens to avoid redundant recomputation.

The memory and computational complexity of this cache scales linearly with sequence length in memory ($\mathcal{O}(N)$) but quadratically in attention computation ($\mathcal{O}(N^2)$), creating a fundamental bottleneck for high-resolution image generation. Our work directly addresses this limitation by exploiting the structured attention patterns that emerge in visual generation.

### 3.2 MARGIN COLUMNS AS SEMANTIC ANCHORS

Our investigation reveals a crucial characteristic of AR visual generation: semantic information from the textual prompt $\mathbf{T}$ distributes non-uniformly across the image token sequence, with pronounced concentration in tokens corresponding to margin columns of the raster-scanned grid. This spatial asymmetry creates natural semantic anchors that orchestrate global image coherence, which informs our compression strategy.

**Semantic Concentration via KV Cache Analysis.** To directly quantify semantic injection patterns, we analyze the divergence between KV caches generated under CFG guidance branch versus non-CFG guidance native branch. Specifically, we compute the Mean Squared Error (MSE) per token position $j$:

$$\mathrm{MSE}_{\mathrm{token}}(j) = \|\mathbf{K}^{\mathrm{cfg}}[:,j] - \mathbf{K}^{\mathrm{native}}[:,j]\|_2^2 + \|\mathbf{V}^{\mathrm{cfg}}[:,j] - \mathbf{V}^{\mathrm{native}}[:,j]\|_2^2. \tag{2}$$

As illustrated in Figure 2(b), empirical analysis reveals that MSE peaks occur at periodic positions corresponding precisely to margin columns in the $24 \times 24$ token grid. These elevated MSE values indicate heightened semantic perturbation in these tokens, confirming their role as privileged conduits for prompt-derived concepts.

This emergent anchoring behavior parallels engineered approaches in models like Lumina-mGPT (Liu et al., 2024a), which explicitly augment margin tokens for adaptive resolution generation. In native AR models, however, this phenomenon arises organically under CFG, revealing an exploitable asymmetry: margin tokens aggregate global semantic information while interior tokens prioritize spatial relationships.

### 3.3 SPATIAL AND SEMANTIC HEAD DICHOTOMY AND THE SSD FRAMEWORK

Guided by our empirical observations, we propose a systematic classification of attention heads into two functionally distinct archetypes:

---

**Algorithm 1** SSD KV Cache Compression

---

**Require:** Model layer $l$, head $i$, current KV cache $\mathbf{K}^i_{<t}$, $\mathbf{V}^i_{<t}$, new token hidden state $\mathbf{h}_t$
**Ensure:** Compressed KV cache $\mathbf{K}^i_{\text{new}}$, $\mathbf{V}^i_{\text{new}}$
 1: **Parameter:** Head type $T_i \in \{\text{Spatial}, \text{Semantic}\}$, window size $W$, memory budget $M$
 2: **Offline:** Classify all heads via sparsity profiling
 3: **if** $T_i = \text{Spatial}$ **then**                              ▷ Apply sliding window policy for spatial heads
 4:     $\mathbf{K}^i_{\text{new}} \leftarrow \text{Append}(\mathbf{K}^i_{<t}[-W :], \mathbf{h}_t \mathbf{W}^i_K)$
 5:     $\mathbf{V}^i_{\text{new}} \leftarrow \text{Append}(\mathbf{V}^i_{<t}[-W :], \mathbf{h}_t \mathbf{W}^i_V)$
 6: **else if** $T_i = \text{Semantic}$ **then**                  ▷ Apply heavy-hitter retention policy for semantic heads
 7:     $\mathbf{a}_t \leftarrow \text{AttentionScores}(\mathbf{h}_t \mathbf{W}^i_Q, \mathbf{K}^i_{<t})$
 8:     $\mathbf{s} \leftarrow \text{AggregateAttentionScores}(\mathbf{a}_{1:t})$                     ▷ Maintain running sum
 9:     $\text{Top}_M \leftarrow \text{IndicesOfTopM}(\mathbf{s}, M)$                              ▷ Select top-$M$ tokens
10:     $\mathbf{K}^i_{\text{new}} \leftarrow \text{SelectAndAppend}(\mathbf{K}^i_{<t}[\text{Top}_M], \mathbf{h}_t \mathbf{W}^i_K)$
11:     $\mathbf{V}^i_{\text{new}} \leftarrow \text{SelectAndAppend}(\mathbf{V}^i_{<t}[\text{Top}_M], \mathbf{h}_t \mathbf{W}^i_V)$
12: **end if**
13: **return** $\mathbf{K}^i_{\text{new}}$, $\mathbf{V}^i_{\text{new}}$

---

- **Semantic Heads**: Characterized by dense, heavy-tailed attention distributions that preferentially attend to global anchors, particularly margin column tokens. These heads maintain overarching semantic coherence and ensure prompt alignment throughout the generation process.

- **Spatial Heads**: Exhibit sparse, localized attention patterns focused primarily on recent neighboring tokens. These heads process spatial relationships and refine local visual details through neighborhood interactions.

**Spatial Sparsity Analysis.** We quantify spatial sparsity patterns by analyzing attention distributions across different layers and heads. For each attention head at layer $l$ and head index $h$, we compute the sparsity metric averaged over multiple generation steps and prompts:

$$s_{l,h} = \frac{1}{P} \sum_{p=1}^{P} \frac{1}{T} \sum_{t=1}^{T} \frac{\sum_{i=0}^{t-1-w} a_{l,h,p,t}(i)}{\sum_{i=0}^{t-1} a_{l,h,p,t}(i)},$$

where $P$ is the number of prompts, $T$ is the max visual token length, $w = 32$ is a recency window, and $a_{l,h,p,t}$ is the attention distribution for specific layer, head, prompt, and timestep. This ratio measures the proportion of attention allocated to tokens beyond the immediate neighborhood, effectively distinguishing globally-attentive heads (high $s$) from locally-focused ones (low $s$). As illustrated in Figure 2(a), Janus-Pro exhibits a structured pyramidal sparsity progression across layers. Lower layers, responsible for processing fundamental visual features (e.g., textures, contours), demonstrate broader attentional spans that establish structural frameworks. In contrast, higher layers progressively focus on local neighborhoods, refining spatial details and fine-grained patterns.

We operationalize this classification using the sparsity metric $s_i$ computed for each head $i$ across diverse prompts and generation steps. The bimodal distribution of $s_i$ values enables robust classification via a threshold $\tau$:

$$\text{HeadType}(i) = \begin{cases} \text{Semantic}, & s_i > \tau \quad \text{(dense, global attention)} \\ \text{Spatial}, & s_i \leq \tau \quad \text{(sparse, local attention)} \end{cases} \tag{3}$$

This functional partitioning transcends mere descriptive categorization, enabling precisely tailored compression strategies that respect the distinct roles of different heads types—a significant advancement over uniform compression approaches derived from language model optimizations.

**SSD Framework.** The Spatial-Semantic (SSD) framework implements head-specific compression policies that leverage the identified functional specialization:

- **Spatial Heads**: Employ *sliding window compression*, retaining only the most recent $W$ tokens along with an initial sink token to anchor global context, inspired by StreamingLLM (Xiao et al., 2023). This strategy capitalizes on the strong locality bias

exhibited by these heads while preserving essential long-range dependencies, safely discarding distant tokens with minimal impact on generation quality.

- **Semantic Heads**: Utilize *heavy-hitter retention*, preserving the top $M$ tokens ranked by accumulated attention scores across generation steps, augmented by a reserved recent window of $R$ tokens. This approach safeguards critical semantic anchors and maintains recency awareness while aggressively compressing less influential tokens.

This asymmetric approach achieves significant efficiency gains—up to $5\times$ memory reduction and $6.6\times$ decoding throughput acceleration—with minimal quality degradation. Algorithm 1 provides the complete implementation.

## 4 EXPERIMENTAL RESULTS

This section presents a comprehensive evaluation of the SSD framework, assessing its effectiveness in preserving multimodal generation quality under aggressive Key-Value (KV) cache compression. We detail the experimental setup, including datasets, comparison methods, and implementation details, followed by quantitative and qualitative results, efficiency metrics, ablation studies, and a discussion of broader implications and future directions.

### 4.1 EXPERIMENT SETTINGS

**Datasets.** We evaluate SSD on two well-established benchmarks designed to test multimodal generation quality and compositional reasoning under varying context complexities: (a) GenEval (Ghosh et al., 2023) features 553 curated prompts assessing compositional reasoning, focusing on object attributes, spatial relationships, and complex understanding. (b) DPG-Bench (Hu et al., 2024) includes 1,065 graph-structured prompts (avg. 84 tokens via CLIP tokenizer) testing prompt adherence and compositional understanding, originally for diffusion but adaptable to autoregressive contexts. These benchmarks span a range of context lengths, enabling evaluation of compression artifacts across fidelity, compositional accuracy, and scalability.

**Comparison Methods.** We compare SSD against two decoding-stage KV cache compression methods and an uncompressed full-cache baseline. (a) StreamingLLM (Xiao et al., 2023) uses a sliding window with sentinel tokens, effective for streaming but with potential long-range dependency issues. (b) H2O (Zhang et al., 2023) dynamically retains high-attention tokens via submodular maximization, improving throughput with minimal perplexity loss. (c) Full Cache: The vanilla transformer decoder with unrestricted KV caching serves as the baseline for fidelity comparisons.

All baseline implementations are sourced from their official repositories and re-tuned for the Janus-Pro model to ensure fair comparisons.

**Implementation Details.** The SSD framework is implemented on the Janus-Pro models (1B and 7B parameters) (Chen et al., 2025), utilizing a $24 \times 24$ token grid (576 tokens) to generate $384 \times 384$ images. For fair comparison across compression methods during quantitative evaluation, we standardize the sliding window size to $W = 32$. Classifier-Free Guidance (CFG) is applied with a guidance scale of $\gamma = 5.0$. Experiments are conducted on NVIDIA A100 GPUs (80GB), with batch sizes up to 256 to assess scalability and robustness.

### 4.2 QUANTITATIVE EVALUATION

We demonstrate that combining window-based locality compression with attention-based semantic compression yields better generation quality than either method alone. We evaluate our framework across two memory scenarios: a low setting (20% token budget) and a high setting (50% token budget).

As shown in Table 1, SSD achieves better GenEval and DPG-Bench scores than H2O and StreamingLLM under 20% and 50% token budgets, comparable to the performance of the vanilla full-cache baseline. These results match our prior observations. Figure 3 further illustrates that SSD maintains robust performance across compression ratios, with minimal degradation even at 20% cache size, aligning with our hypothesis that combining window-based and attention-based compression leverages both local and global dependencies effectively.

Table 1: Performance on GenEval and DPG-Bench under 20% and 50% cache sizes, compared to Full Cache, StreamingLLM, and H2O. Our Method consistently achieves comparable performance to Full Cache with significantly reduced memory.

| Method | Cache Size | GenEval | | | | DPG | | | |
|---|---|---|---|---|---|---|---|---|---|
| | | Two Obj. | Counting | Color Attri. | Overall↑ | Entity | Attribute | Relation | Overall↑ |
| **Janus-Pro-1B** | | | | | | | | | |
| Full | 100% | 0.82 | 0.51 | 0.56 | 0.73 | 89.06 | 89.18 | 89.73 | 82.98 |
| Streaming | | 0.77 | 0.28 | 0.44 | 0.64 | 87.28 | 87.60 | 87.78 | 81.01 |
| H2O | 20% | 0.74 | 0.44 | 0.43 | 0.67 | 87.49 | 87.54 | 85.48 | 80.44 |
| Ours | | **0.81** | **0.48** | **0.57** | **0.73** | **89.23** | **87.94** | **89.48** | **82.82** |
| Streaming | | 0.80 | 0.48 | 0.55 | 0.72 | **89.75** | 88.67 | **89.92** | 83.17 |
| H2O | 50% | **0.81** | 0.49 | 0.56 | 0.67 | 88.96 | 88.47 | 86.58 | 82.28 |
| Ours | | **0.81** | **0.52** | **0.58** | **0.74** | 88.54 | **89.31** | 88.90 | **83.21** |
| **Janus-Pro-7B** | | | | | | | | | |
| Full | 100% | 0.89 | 0.59 | 0.66 | 0.80 | 89.35 | 90.24 | 90.45 | 84.77 |
| Streaming | | **0.87** | 0.52 | 0.62 | 0.77 | 86.45 | 87.51 | 89.48 | 83.20 |
| H2O | 20% | 0.78 | 0.53 | 0.47 | 0.71 | 81.90 | 83.86 | 83.12 | 75.97 |
| Ours | | 0.86 | **0.58** | **0.67** | **0.79** | **89.52** | **89.26** | **90.17** | **84.45** |
| Streaming | | 0.87 | 0.59 | 0.63 | 0.79 | 88.70 | 88.89 | **92.22** | **84.83** |
| H2O | 50% | 0.86 | **0.60** | 0.62 | 0.79 | 87.37 | 87.73 | 89.47 | 83.36 |
| Ours | | **0.89** | **0.60** | **0.65** | **0.80** | **88.92** | **89.51** | 89.19 | 84.72 |

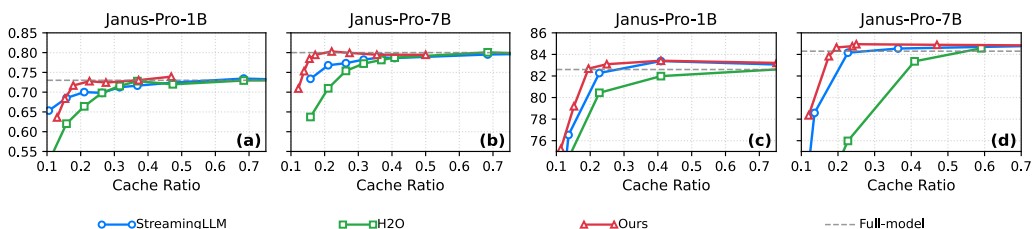

Figure 3: Performance comparison of SSD, Full Cache, StreamingLLM, and H2O on (a,b) GenEval and (c,d) DPG-Bench across varying compression ratios for Janus-Pro-1B and Janus-Pro-7B. SSD achieves performance comparable to Full Cache while significantly reducing memory usage, outperforming both H2O and StreamingLLM on both benchmarks.

## 4.3 QUALITY EVALUATION

To qualitatively assess the detailed impact of KV cache compression, we carefully visualize generated images from Janus-Pro under various compression ratios (20%, 50%) in Figure 4. The examples highlight the ability of SSD to consistently preserve fine details, spatial relationships, and semantic coherence even under aggressive compression. These visuals conclusively corroborate the quantitative gains, demonstrating the robustness of SSD in real-world generation tasks.

## 4.4 EFFICIENCY EVALUATION

We assess the effectiveness of SSD in reducing memory consumption and enhancing time efficiency during inference by analyzing peak memory usage and throughput across different batch sizes.

**Peak Memory Usage.** As depicted in the left panel of Table 3, SSD exhibits substantial memory savings capabilities, comparable to other KV cache eviction methods, both attention-based (e.g., H2O) and window-based (e.g., StreamingLLM). All these approaches maintain a fixed-size KV cache. When compared to the full cache implementation, SSD achieves an impressive reduction in peak memory usage of up to 5× with batch size of 256.

**Throughput.** As batch size increases, decoding latency for the full cache method grows significantly due to escalating computational demands and I/O latency bottlenecks. In contrast, SSD maintains a relatively stable throughput by preserving a fixed amount of KV cache, resulting in significantly lower latency compared to the full cache, particularly for larger batch sizes. It is noteworthy that

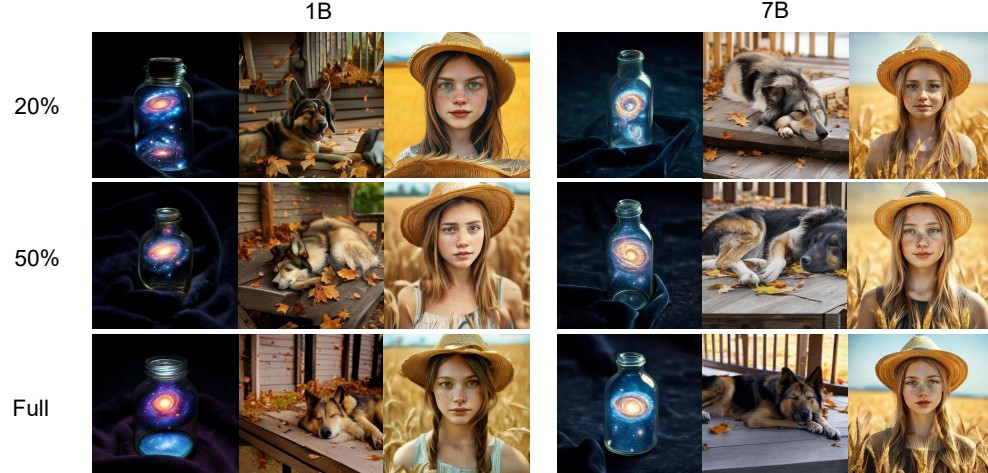

(a) "A tiny galaxy contained inside a glass bottle, glowing brightly sagainst a dark velvet cloth."

(b) "A shepherd dog lying peacefully on a wooden porch, with autumn leaves scattered around."

(c) "A young woman with freckles wearing a straw hat, standing in a golden wheat field."

Figure 4: Qualitative visualization of generated images from Janus-Pro-1B and Janus-Pro-7B under varying compression ratios (20% to 50% cache sizes) for SSD and Full Cache. SSD preserves fine details, spatial relationships, and semantic coherence in complex scene generation.

SSD demonstrates remarkable efficiency, achieving over $6.6\times$ speedup in throughput compared to the full cache approach when processing batches of 128.

## 4.5 Ablation Study

To further validate the key components of SSD, we conduct ablation experiments on the Janus-Pro-7B model. Our analysis focuses on two critical aspects: head selection strategy and threshold sensitivity.

**Threshold Sensitivity Analysis.** We investigate the impact of threshold selection $\tau$ on dividing the semantic KV cache. By varying $\tau$ across (0, 0.45, 0.9) as shown in Figure 5, we observe that higher thresholds progressively filter heads to those with more pronounced semantic concentration. Specifically, higher threshold $\tau$ exhibits higher spiked MSE patterns at margin columns, confirming that global semantic information predominantly resides in heads whose sparsity $> \tau$, which validates our threshold-based approach for identifying semantic-critical heads.

Table 2: Quantitatively comparison between randomly partition and our method.

| Methods | Geneval | DPG-Bench |
|---------|---------|-----------|
| Random | 0.75 | 81.23 |
| Ours | 0.79 | 84.65 |
| Baseline | 0.80 | 84.19 |

**Effectiveness of Head Selection Strategy.** We further evaluate SSD against random head assignment mimicking our asymmetric policy allocation but without semantic justification. As shown in Table 2, our sparsity-based dichotomy ($\tau = 0.8$) achieves better GenEval and DPG-Bench evaluation scores, demonstrating that distinguishing semantic and spatial heads is crucial for effective asymmetric compression. The performance gap underscores the importance of our head classification over heuristic alternatives.

## 4.6 Discussion and Future Work

We argue that our contributions establish that effective KV cache compression for AR image generation necessitates a hybrid approach combining window-based locality compression and attention-aware semantic retention. This strategy adeptly addresses the spatial-semantic dichotomy inherent in visual token generation, outperforming language-derived methods that overlook visual-specific patterns (Zhang et al., 2023; Xiao et al., 2023).

**Computational Overhead Mitigation.** The dynamic token identification process in the SSD framework introduces moderate computational and memory allocation overhead due to iterative attention

Table 3: System performance scaling with batch size. SSD demonstrates superior efficiency compared to full cache, achieving up to 6.6× throughput improvement and 5× memory reduction, which highlights its practical advantage in real-world deployment scenarios.

| BS | Throughput (tokens/s) | | Memory (GB) | |
|----|------|------|------|------|
| | Full | SSD | Full | Ours(SSD) |
| 32 | 897.3 | 1253.0 (1.4×) | 18.1 | 5.9 |
| 64 | 543.6 | 1658.4 (3.1×) | 32.2 | 7.9 |
| 128 | 289.2 | 1911.7 (6.6×) | 60.4 | 11.8 |
| 256 | OOM | 2037.4 | OOM | 19.7 |

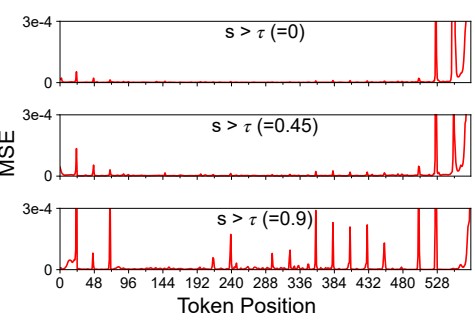

Figure 5: Threshold sensitivity analysis of $\tau$ on Janus-Pro-7B, higher $\tau$ values exhibiting pronounced MSE spikes at margin columns.

score calculations. To address this, we propose a buffering mechanism that stores candidate tokens in a temporary cache, updated periodically to minimize redundant computations and resource usage. Preliminary experiments, as shown in Table 4, demonstrate that this optimization reduces computational overhead by 50% while preserving generation quality, with negligible impact on GenEval and DPG-Bench metrics. Further details are available in Appendix D.

Table 4: Throughput and memory efficiency across batch sizes with 20% cache. SSD with buffer method delivers significant performance improvements—achieving up to 10.7× higher throughput than full cache—while reducing memory consumption by approximately 80%. Notably, these efficiency gains are achieved without substantial degradation in task performance, as evidenced by comparable scores on GenEval and DPG evaluation benchmarks.

| Method | BS = 32 | | BS = 64 | | BS = 128 | | GenEval | DPG |
|--------|---------|------|---------|------|----------|------|---------|-----|
| | Thr. | Mem. | Thr. | Mem. | Thr. | Mem. | | |
| Full Cache | 897.3 | 18.1 | 543.6 | 32.2 | 289.2 | 60.4 | 0.73 | 82.63 |
| SSD | 1253.0 (1.4×) | 5.9 | 1658.4 (3.1×) | 7.9 | 1911.7 (6.6×) | 11.8 | 0.73 | 82.82 |
| SSD w/buffer | **1754.1** (2.0×) | 5.8 | **2524.6** (4.6×) | 7.7 | **3099.4** (10.7×) | 11.6 | 0.72 | 82.53 |

**Integration with Other Compression Techniques** The text modeling field provides extensive research on prefill-phase compression, including layer-wise (Cai et al., 2024) and head-wise (Feng et al., 2024) allocation strategies. These methods, tailored to enhance the initial prompt processing stage, can be effectively integrated with the decoding-phase optimization of SSD. Future research could investigate hybrid frameworks that merge layer-wise or head-wise budget allocation compression with our adaptive decoding policies to enhance overall efficiency.

## 5 CONCLUSION

Unified autoregressive models for image generation face significant and increasingly critical memory challenges due to the exceptionally large number of visual tokens processed during inference. While KV cache eviction methods have greatly improved memory efficiency in text-based models, their application to visual autoregressive generation has been largely underexplored.

We address this gap by identifying distinct spatial and semantic attention patterns in visual AR models and proposing SSD, a novel compression framework that applies tailored policies to spatial and semantic attention heads. This approach achieves substantial memory reduction and inference speedup while preserving high-quality output. To our knowledge, SSD is the first comprehensive framework for KV cache compression in unified visual AR models, paving the way for efficient deployment of large-scale multimodal systems and advancing practical inference solutions.

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

## A  DECLARATION OF LLM USAGE

We use the Large Language Models (LLMs) for this paper to serve one purpose: to aid and polish the paper writing. We use the LLMs in a very limited capacity, restricted to minor editing of grammar, phrasing, and readability. We do not involve the LLMs in designing the method, developing theoretical results, and conducting experiments.

## B  GENERALIZABILITY

We further analyze the attention sparsity distribution across all heads in LlamaGen-XL. Some heads exhibit extremely high sparsity, while the remaining heads stay relatively dense, confirming the strong generalizability of our method.

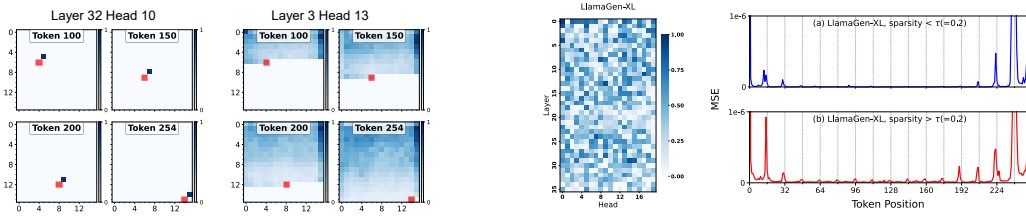

Figure A1: Left: Visualization of the reshaped attention across different generation steps in LlamaGen-XL; Right: attention sparsity patterns and token-wise MSE between conditional and CFG-augmented branches.

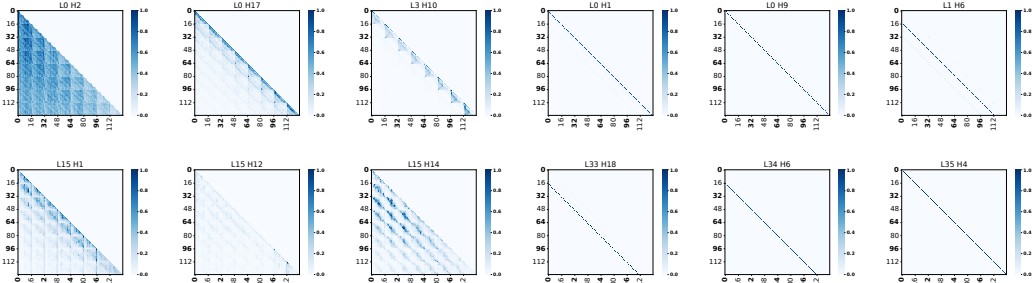

Figure A2: More attention visualization in LlamaGen

## C  MORE RELATED WORKS

Prior works have extensively explored spatial locality in autoregressive (AR) visual generation to accelerate decoding. For instance, Locality-aware Parallel Decoding (LPD) (Zhang et al., 2025b) leverages local spatial dependencies for parallel token prediction, reducing inference latency in high-resolution image synthesis. Similarly, Neighboring Autoregressive Modeling (NAR) (He et al., 2025b) models visual tokens with neighborhood-aware autoregression, enabling efficient generation by focusing on proximate spatial relationships. ZipAR (He et al., 2025a) further advances this by exploiting spatial locality for parallel AR decoding, achieving significant speedups without sacrificing fidelity. These methods optimize generation parallelism via locality exploitation, often requiring architectural changes or training, and are orthogonal to our SSD framework, which applies locality insights to KV cache compression in standard AR decoders with training-free.

Attention sink phenomena, where certain tokens aggregate disproportionate attention, have been observed since the Vision Transformer (ViT) era. Works like "See What You Are Told" (Kang et al., 2025) analyze visual attention sinks in large multimodal models, showing how they stabilize generation by anchoring semantic information. Quantizable Transformers (Bondarenko et al., 2023) identify outlier attention heads that "do nothing" effectively, enabling quantization by pruning redundant

sinks. In language models, MInference 1.0 (Jiang et al., 2024) accelerates long-context pre-filling via dynamic sparse attention, leveraging sink-like patterns for efficient KV handling. While these establish the prevalence of sinks in vision and language, our work extends this by identifying emergent semantic sinks in native AR image models under CFG, where semantic information aggregates in margin columns. We further demonstrate that this leads to head decoupling for differentiated compression (e.g., window for spatial heads, heavy-hitter for sink heads), outperforming uniform strategies like those in LLMs.

# D  BUFFER MECHANISM

This appendix evaluates an enhanced buffering mechanism in the SSD framework to boost inference throughput while preserving high-quality image generation in the Janus-Pro-1B model. The buffering mechanism stores recent tokens in a temporary cache, refreshed every 24 steps, to reduce redundant attention computations. The detailed algorithm is presented in Algorithm 2.

---

**Algorithm 2** SSD KV Cache Compression with Row Buffer

---

**Require:** Model layer $l$, head $i$, new token hidden state $\mathbf{h}_t$, key $\mathbf{k}_t$, value $\mathbf{v}_t$
**Ensure:** Compressed KV cache $\mathbf{K}_{\text{new}}^i$, $\mathbf{V}_{\text{new}}^i$
 1: **Parameters:** Head type $T_i \in \{\text{Spatial}, \text{Semantic}\}$, window size $W$, memory budget $M$, row buffer size $R$
 2: **Offline:** Classify heads via importance profiling
 3: **if** Prefilling stage (num_new_tokens > 1) **then**
 4:     Split $\mathbf{k}_t, \mathbf{v}_t$ into spatial and semantic parts; Return for attention
 5: **else if** Decoding stage (num_new_tokens == 1) **then**
 6:     **if** Row buffer index + 1 ≤ $R$ **then**
 7:         Append $\mathbf{k}_t, \mathbf{v}_t$ to row buffer; Increment row buffer index
 8:         **if** Row buffer index == $R$ **then**
 9:             Set row buffer full flag
10:         **end if**
11:         Split row buffer KV into spatial and semantic
12:         $\mathbf{K}_{\text{spatial}}, \mathbf{V}_{\text{spatial}} \leftarrow$ Concat(cached spatial KV, buffer KV)
13:         $\mathbf{K}_{\text{semantic}}, \mathbf{V}_{\text{semantic}} \leftarrow$ Concat(cached semantic KV, buffer KV)
14:         $\mathbf{K}_{\text{new}}^i, \mathbf{V}_{\text{new}}^i \leftarrow$ Concat(spatial KV, semantic KV)
15:     **else**
16:         Raise error: Row buffer overflow
17:     **end if**
18: **end if**
19: **Post-Process:**
20: **if** Attention scores available **then**
21:     Accumulate semantic attention scores
22: **end if**
23: **if** Row buffer full or prefill **then**
24:     Compress spatial cache (sink + recent); Compress semantic cache (heavy-hitter)
25:     Reset row buffer
26: **end if**
27: **return** $\mathbf{K}_{\text{new}}^i, \mathbf{V}_{\text{new}}^i$

---

# E  MORE INFORMATION ABOUT MODEL AND SPARSITY

We provide the model architecture configurations and sparsity distribution across heads in Table A1 and Table A2.

Table A1: Model specifications for Janus-Pro

| Model | Image Tokens | Layers | Embedding Size | Attention Heads |
|-------|--------------|--------|----------------|-----------------|
| Janus-Pro-1B | 576 | 24 | 2048 | 16 |
| Janus-Pro-7B | 576 | 30 | 4096 | 32 |

Table A2: Sparsity distribution across heads in Janus-Pro models

| Sparsity Range | Percentage (%) | | Count | | Cumulative % | | Cumulative Count | |
|----------------|-----|-----|-----|-----|-----|-----|-----|-----|
| | 1B | 7B | 1B | 7B | 1B | 7B | 1B | 7B |
| 0.0–0.1 | 34.1 | 21.8 | 131 | 209 | 34.1 | 21.8 | 131 | 209 |
| 0.1–0.2 | 16.4 | 16.6 | 63 | 159 | 50.5 | 38.3 | 194 | 368 |
| 0.2–0.3 | 12.2 | 10.2 | 47 | 98 | 62.8 | 48.5 | 241 | 466 |
| 0.3–0.4 | 9.4 | 8.8 | 36 | 84 | 72.1 | 57.3 | 277 | 550 |
| 0.4–0.5 | 7.8 | 11.0 | 30 | 106 | 79.9 | 68.3 | 307 | 656 |
| 0.5–0.6 | 8.9 | 8.6 | 34 | 83 | 88.8 | 77.0 | 341 | 739 |
| 0.6–0.7 | 5.2 | 8.1 | 20 | 78 | 94.0 | 85.1 | 361 | 817 |
| 0.7–0.8 | 2.6 | 8.8 | 10 | 84 | 96.6 | 93.9 | 371 | 901 |
| 0.8–0.9 | 1.6 | 3.6 | 6 | 35 | 98.2 | 97.5 | 377 | 936 |
| 0.9–1.0 | 1.8 | 2.5 | 7 | 24 | 100.0 | 100.0 | 384 | 960 |

## F  MORE ATTENTION VISUALIZATION

We further analyze the attention sparsity distribution across all heads in LlamaGen-XL. As shown in the right panel, a subset of heads exhibits extremely high sparsity ($>0.9$), while others remain relatively dense.

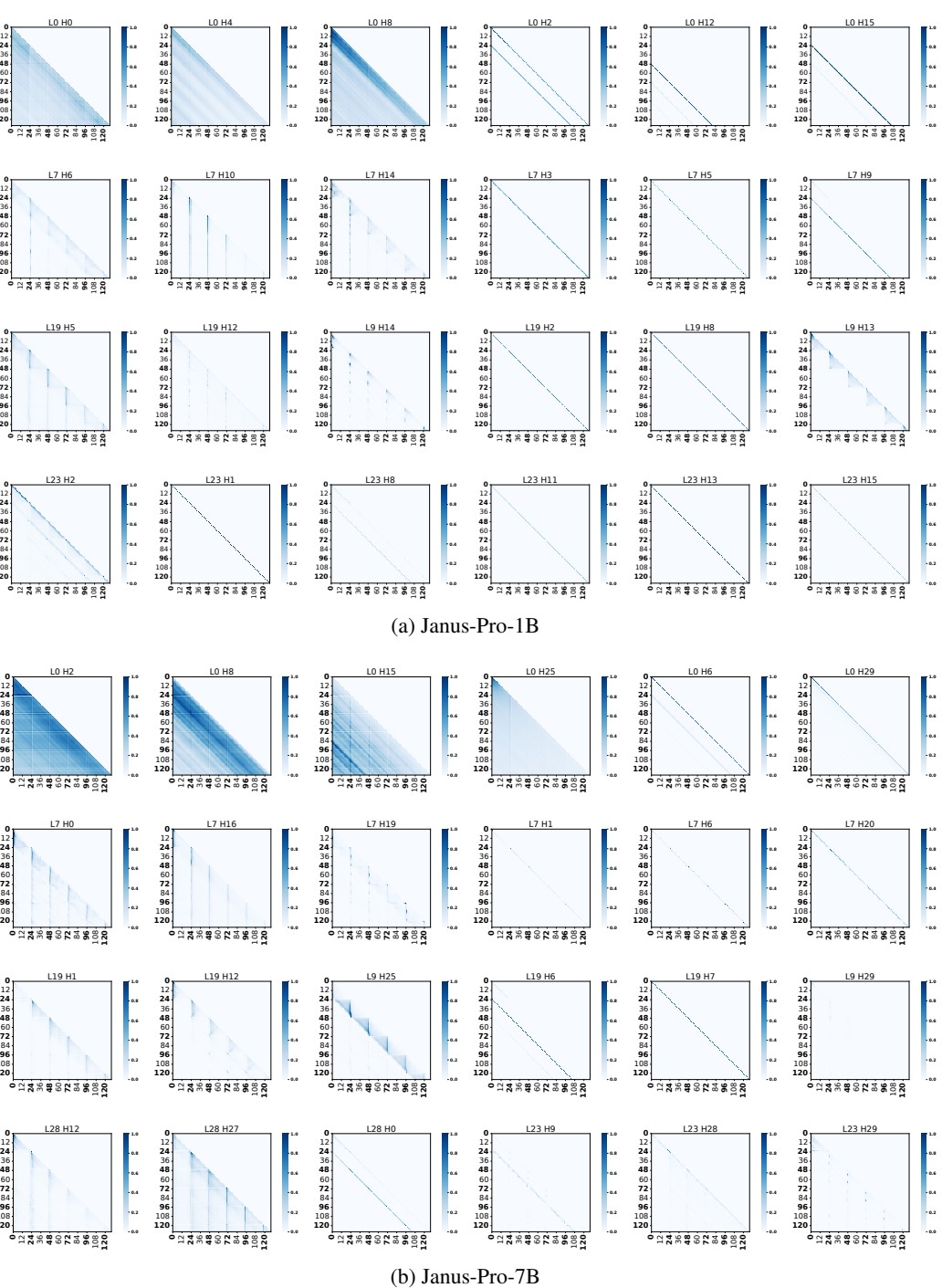

(a) Janus-Pro-1B

(b) Janus-Pro-7B

Figure A3: More attention visualization

