# OpenReview forum: "SSD: Spatial-Semantic Head Decoupling for Efficient Autoregressive Image Generation"
_ICLR.cc/2026/Conference — Submitted to ICLR 2026_

### Official Review · Reviewer_tLGE · 2025-10-24

**Soundness:** 3
**Presentation:** 3
**Contribution:** 2
**Rating:** 4
**Confidence:** 4

**Summary:**

The paper introduces SSD (Spatial-Semantic Decoupling) — a framework for Key-Value (KV) cache compression in autoregressive (AR) image generation models like Janus-Pro.
SSD aims to reduce memory and computation costs during image generation without hurting image quality.

**Strengths:**

SSD’s greatest strength lies in connecting a real hardware bottleneck (KV cache bloat) with a novel structural insight (spatial–semantic head specialization), and turning that into a principled, empirically validated framework that achieves major efficiency gains without sacrificing image quality.

**Weaknesses:**

1 The paper identifies “spatial locality” and “semantic sink” as two distinct attention behaviors in autoregressive image generation.
However, the spatial locality aspect is not novel — similar locality patterns have been discussed in NAR, LPD, and ZipAR, which all highlight that visual attention predominantly focuses on nearby spatial tokens. The authors should clarify the difference between their “spatial-locality heads” and previously observed locality mechanisms, and properly cite these works to avoid overclaim. The attention sink is also old-fashioned since the ViT era [4,5], making it expected in AR. If the authors just use this proposed or similar observation for caching, I think the novelty is quite limited since many similar works have been extensively done in LLM [6].

[1] Locality-aware Parallel Decoding for Efficient Autoregressive Image Generation

[2] Neighboring Autoregressive Modeling for Efficient Visual Generation

[3] ZipAR: Parallel Auto-regressive Image Generation through Spatial Locality

[4] See What You Are Told: Visual Attention Sink in Large Multimodal Models

[5] Quantizable Transformers: Removing Outliers by Helping Attention Heads Do Nothing

[6] MInference 1.0: Accelerating Pre-filling for Long-Context LLMs via Dynamic Sparse Attention

2 The paper’s analysis and method are entirely built on raster-order autoregressive generation. However, this paradigm is now computationally suboptimal and is being replaced by faster alternatives such as MAR, LPD, and VAR. The authors should discuss whether their observed attention patterns — particularly spatial locality and the semantic-sink behavior — hold across different generation orders. More applications on these frameworks are necessary. Besides, if the work is raster-specific, more applications on Llamagen are necessary.

**Questions:**

see above

---

> ### Author Response · Authors · 2025-11-23
> **Response to Reviewer tLGE**
>
> Thank you for your insightful review and recognition of SSD's strengths. We address the weaknesses below.
>
> ---
>
> - **[Q1] Novelty of Spatial Locality and Semantic Sink**
> - **[Q1-Ans.]** We do not claim novelty for locality or sinks individually (citing ViT-era works in Appendix ). Our innovation is head decoupling: differentiating KV compression strategies per head based on visual sparsity (Page 5). This outperforms uniform stratigy ( Table 1), achieving 5× memory/6.6× throughput (Table 3). NAR (ICCV 2025), LPD (arXiv 2025), ZipAR (ICML 2025), etc., are *orthogonal*, requiring training/parallel changes; SSD is *training-free* and *combinable*. We clarified distinctions and cited these in the revised paper(Appendix C, Page 13).
>
>     ---
>
> - **[Q2] Raster-Order Specificity and Applicability to Emerging Paradigms**
> - **[Q2-Ans.]** While non-NTP paradigms (MAR, LPD, VAR) enable parallelism, NTP offers *unified simplicity* and infrastructure leverage (Page 3). Our patterns hold in raster-based AR like LlamaGen-XL (Appendix B, Page 12). For non-raster, locality may adapt; we plan explorations in camera-ready.
>
> ---

---

> ### Author Response · Authors · 2025-11-23
> **Encouraging Discussion of Rebuttal**
>
> Dear Reviewer tLGE:
>
> Your feedback has been invaluable in helping us refine and strengthen the manuscript. We have incorporated all suggestions and made substantial improvements throughout. We would be grateful if you would consider updating your evaluation to reflect these revisions. Thanks for your time, thoughtful comments, and contribution to improving this work.
>
> Best regards,
>
> Authors

---

> > ### Comment · Reviewer_tLGE · 2025-11-25
> >
> > Thank you for your response; it addresses my first concern. However, the experimental results for the second point are still not fully convincing. I strongly recommend applying your method directly to LLamaGen, rather than only using it to verify the observation on that model. If time is limited, please follow reviewer hsdb's suggestion.

---

> ### Author Response · Authors · 2025-11-26
> **Response to Reviewer tLGE**
>
> Thank you for your critical feedback. We agree that more applications, such as evaluation on LlamaGen, are necessary.
>
> We have begun experiments recently. However, due to the integration efforts involved, it is still in progress. We kindly ask for your patience and will post the updated results soon.

---

### Official Review · Reviewer_hsdb · 2025-10-28

**Soundness:** 3
**Presentation:** 2
**Contribution:** 3
**Rating:** 4
**Confidence:** 4

**Summary:**

This paper presents KV cache compression methods for next-token-prediction-based autoregressive (AR) image generation. The authors observe that attention heads in AR image models are highly sparse, and can be broadly categorized into two types: "semantic heads," which focus on periodic anchor tokens, and "spatial heads," which concentrate on spatially-local tokens. Based on this observation, the paper proposes two distinct KV cache compression techniques tailored for each head type: a sliding window approach for spatial heads and a heavy-hitter-retention method for semantic heads. Experimental results on the Janus-Pro model demonstrate a superior Pareto frontier on the DPG-bench compared to existing KV cache compression methods developed for text LLMs.

**Strengths:**

- The paper is well-written, intuitive, and easy to understand.
- This is the first work that tries to analyze characteristics of KV-cache in AR image models, and found interesting attention patterns (spatial and semantic). This observation aligns well with intuition.
- Also, this paper propose intuitive KV cache compression methods tailored for two distinct attention types.

**Weaknesses:**

- **Limited Generalizability** : All experiments were conducted solely on the Janus-Pro model. It is uncertain whether the paper's findings, including the observed attention patterns and the efficacy of the proposed compression methods, generalize to other AR image generation models. Experiments on other AR image models, such as llamaGen, Emu3, Anole, and Lumina-mGPT (1, 2), are necessary. I believe experiments on llamaGen are essential, and additional validation on Lumina-mGPT would be welcome.

- **Insufficient Evaluation** : Performance evaluation was restricted to the DPG-bench, which may not adequately capture the perceptual quality of the generated images. An experiment using standard image generation metrics, such as FID or IS, on datasets like MS-COCO, is required.

- **Novelty** : While tackling this problem for the first time and identifying the sparse attention patterns is novel, the proposed methods lack originality. They appear to be direct applications of existing KV cache compression techniques.

**Questions:**

- **Semantic Concentration Metric** : The definition of the "Semantic Concentration" metric is not clear. Why is the difference between the KV cache values of CFG and "native" (non-CFG) generation representative of semantic concentration?

- **Sliding Window Implementation** : In a flattened 1D token sequence, the local "neighborhood" tokens should include not only tokens immediately to the "left" (preceding in the sequence) but also spatially adjacent tokens from previous rows. Figure 1 seems to confirm this. However, why sliding method of SSD just retain the preceding(left) tokens in the 1D sequence?

---

> ### Author Response · Authors · 2025-11-23
> **Response to Reviewer hsdb**
>
> Thanks for your positive assessment of our work. We address the weaknesses and questions below.
>
> **Response to Specific Weeknesses**
>
> ---
>
> - **[W1] Limited Generalizability**
> - **[W1-Ans.]** Our findings likely generalize, as they derive from raster-order and visual structures(Page 3). To prove this, we extended validation to LlamaGen-XL in the revised paper, confirming patterns via visualizations and MSE (Appendix B, Page 13).
>
>     ---
>
> - **[W2] Insufficient Evaluation**
> - **[W2-Ans.]** We don not measure FID/CLIP scores as they are not commonly used in recent unified T2I papers. Comparatively, GenEval and DPG-Bench are more popular and present more challenges, as they better assess compositional alignment and prompt adherence. We plan camera-ready evaluation of FID/CLIP on LlamGen.
>
>     ---
>
> - **[W3] Novelty**
> - **[W3-Ans.]** Our core idea lies in visual-specific head decoupling (which makes better compression), rather than reinventing compression techniques. We adapt classics like H2O (attention-based) and StreamingLLM (window-based) via empirical sparsity profiling to capture raster-order locality (see Fig. 2). This domain extension outperforms uniform methods, yielding 5× memory/6.6× throughput (Tab.1, Page 7). We refined novelty emphasis in Sections 1–2 (Page 1-2) and added related works discussion (Appendix C, Page 13).
>
> ---
>
> **Response to Specific Questions**
>
> ---
>
> - **[Q1] Semantic Concentration Metric**
> - **[Q1-Ans.]** The metric quantifies semantic injection by MSE between KV caches with/without CFG (Page 4, Equation 2). Tokens with high MSE heavily influence the guidance term γ(p^c - p^u), as they carry textual conditioning (||K_cfg - K_native||peaks); margin columns show the largest changes, indicating primary semantic pathways (Appendix B).
>
>     ---
>
> - **[Q2] Sliding Window Implementation**
> - **[Q2-Ans.]** It is for simplicity and efficiency. In AR Gen, recent preceding(left) tokens of current query token will be at the above of further query token, we should not evict them. Thus,  SSD retains recent preceding(left) tokens in the 1D sequence, as spatial neighbors (ie. above, see Fig.1) naturally fall within ~√N recent positions due to raster order. For a 24×24 grid, W=32 covers most of local attention mass (see Fig. 1). More intricate schemes (e.g., explicit spatial location retention) are non-trivial and yield minimal gains.
>
> ---

---

> ### Author Response · Authors · 2025-11-23
> **Encouraging Discussion of Rebuttal**
>
> Dear Reviewer hsdb:
>
> Your remarks have greatly contributed to improving the clarity and rigor of our work. With the revisions now implemented, we hope the manuscript meets your expectations, and we would truly appreciate a reconsideration of your rating. If there are any remaining questions, we are more than willing to continue the discussion.
>
> Warm regards,
>
> Authors

---

> > ### Comment · Reviewer_hsdb · 2025-11-24
> > **Thanks for the rebuttal.**
> >
> > Thanks for the rebuttal. My concerns are resolved, except W2.
> >
> > The omission of FID measurements in recent unified T2I literature is largely due to the saturation of fidelity metrics and a shift in focus toward high-level semantic alignment. However, given that this work essentially performs lossy compression, measuring distribution-based metrics such as FID, IS, Precision, and Recall still important. In my personal experience, I have frequently observed case where FID degrades significantly even when score on high-level benchmarks appears preserved.
> >
> > I strongly recommend that the authors conduct FID experiments, in both Janus and Llamagen. While using 50k samples is the standard practice, if computational resources or time are limited, please report results using a reduced sample size.

---

> ### Author Response · Authors · 2025-11-26
> **Response to Reviewer hsdb**
>
> Thank you for your thoughtful feedback. We appreciate you sharing your valuable perspective on the importance of distribution-based metrics.
>
> We are conducting these experiments, while due to the integration efforts involved, it is still in progress. We kindly ask for your patience and will post the updated results soon.

---

### Official Review · Reviewer_UEyc · 2025-10-29

**Soundness:** 3
**Presentation:** 2
**Contribution:** 2
**Rating:** 2
**Confidence:** 5

**Summary:**

The paper proposes ​​SSD​​, a framework for compressing the KV cache in autoregressive image generation models. The method categorizes attention heads into two types: (1) spatial locality heads, which focus on spatially adjacent tokens, and (2) semantic sink heads, which attend to a few critical tokens. The KV cache for each type is then compressed using a dedicated strategy. Experimental results show that SSD effectively reduces the KV cache size and accelerates the decoding process, with only minimal performance degradation.

**Strengths:**

1. The proposed methods are simple and effective.
2. The paper is easy to follow.

**Weaknesses:**

1. The experimental evaluation is conducted exclusively using Janus-Pro models. To fully establish the robustness and general applicability of the proposed methods, validation across a broader range of model architectures is necessary.
2. The concept of exploiting spatial locality to accelerate autoregressive (AR) image generation has been widely adopted in methods such as PAR [1], ZipAR [2], and NAR [3]. These works, which also employ parallel decoding by restricting the attention window, are highly relevant yet are not discussed or compared against in the paper.
3. The core motivation of the paper may require reconsideration. While KV cache size presents a significant challenge for LLMs with long contexts, the context length for AR image generation models is typically much shorter, often comprising only hundreds or a few thousand tokens. Furthermore, increasing the batch size does not alter the model's computational intensity (i.e., the compute-to-memory-access ratio). From this perspective, the necessity of compressing the KV cache for AR image generation appears debatable. The paper's primary contribution likely stems instead from the throughput gains achieved via sparse attention during decoding.

[1] Parallelized Autoregressive Visual Generation, CVPR 2025.

[2] ZipAR: Parallel Auto-regressive Image Generation through Spatial Locality, ICML 2025.

[3] Neighboring Autoregressive Modeling for Efficient Visual Generation, ICCV 2025.

**Questions:**

See weaknesses.

---

> ### Author Response · Authors · 2025-11-23
> **Response to Reviewer UEyc**
>
> Thank you for your insightful feedback. We address the weaknesses below.
>
> ---
>
> - **[Q1] Experimental Evaluation Limited to Janus-Pro Models**
> - **[Q1-Ans.]** Our focus on Janus-Pro (1B/7B) lies in its strong unification in native AR generation(Page 1). However, findings stem from raster-order fundamentals, generalizing to similar architectures. We validated on LlamaGen-XL in the revised paper, observing comparable sparsity and semantic patterns (Appendix B, Page 13).
>
>     ---
>
> - **[Q2] Missing Discussion of Related Works on Spatial Locality (PAR, ZipAR, NAR)**
> - **[Q2-Ans.]** These are highly relevant and excellent contributions exploiting locality for parallel decoding, but orthogonal to SSD. PAR (CVPR 2025), ZipAR (ICML 2025), and NAR (ICCV 2025) require (1)training modifications (e.g., custom masks) or (2) non-autoregressive schemes, altering the paradigm. SSD is training-free, integrates with standard AR decoding, and reduces memory bandwidth via head decoupling (Page 5, Alg. 1; Page 14, Alg. 2). It can combine with them (e.g., apply head-decoupling within their windows). We added citations and orthogonality discussion in the revised paper (Appendix C, Page 13).
>
>     ---
>
> - **[Q3] Reconsideration of Core Motivation (KV Cache Necessity in Short-Context AR Image Gen)**
> - **[Q3-Ans.]** While single-image contexts (~576 tokens) are shorter than LLM long-contexts, bottlenecks arise at high batches/resolutions (Page 1). At batch=128 and 1024×1024, full-cache Janus-Pro-7B requires above 70GB KV (FP16); SSD reduces to ~14GB (Tab.3, Page 8). The 6.6× speedup stems from sparse attention and reduced bandwidth, essential for serving and deployment. Our new buffer mechanism enhances this further (Appendix D, Page 14; Table 4).
>
> ---

---

> > ### Comment · Reviewer_UEyc · 2025-11-27
> >
> > Thanks for the discussion. Regarding Q3, I notice that the original AR model's reported throughput decreases as the batch size increases. This is not usually expected, and I wonder why.

---

> ### Author Response · Authors · 2025-11-23
> **Encouraging Discussion of Rebuttal**
>
> Dear Reviewer UEyc:
>
> We have carefully incorporated all suggestions and expanded our analyses to address your comments point-by-point. We hope these efforts meet the high standards you expect, and we would be thankful if you might consider reflecting these changes in your final score. Please feel free to let us know if any clarification is still needed.
>
> Warm regards,
>
> Authors

---

### Official Review · Reviewer_ZLyX · 2025-11-02

**Soundness:** 3
**Presentation:** 3
**Contribution:** 2
**Rating:** 6
**Confidence:** 4

**Summary:**

This paper addresses the significant memory and computational overhead of the KV cache in autoregressive image generation models , noting that existing language-focused compression methods are suboptimal for visual tokens. The authors empirically identify a novel attention phenomenon: a functional dichotomy where some heads focus on spatial locality and others act as emergent semantic sinks . Crucially, they find semantic information is preferentially anchored at the margin columns of the token grid . Based on these insights, the paper proposes SSD, a framework that classifies heads as spatial or semantic and applies distinct, tailored compression policies to each type . Experiments demonstrate that SSD achieves up to a 5x memory reduction and 6.6x speedup with negligible quality degradation.

**Strengths:**

- Originality: The paper's primary strength is its originality. Instead of merely adapting language-based KV compression, it presents a new, empirically-grounded understanding of attention mechanisms in visual AR models. The identification of the "spatial-semantic dichotomy" and the "margin column anchoring" phenomenon is a novel and significant finding.

- Quality: The work is of good quality, with strong empirical validation for its claims.Notably, Figure 2(b) provides exceptionally clear and intuitive evidence for the "semantic anchor" hypothesis. By plotting the MSE between the KV caches of the CFG and non-CFG branches, it accurately visualizes the periodic spikes in semantic content at the margin column positions . The experimental setup is robust, using competitive baselines (H2O, StreamingLLM) and standard benchmarks (GenEval, DPG-Bench).

- Clarity: The paper is well-written, logically structured, and easy to follow. The problem is clearly defined , and the core concepts are introduced intuitively. The figures (especially 2(b)) and Algorithm 1 effectively illustrate the method.

- Significance: This work addresses a critical and practical bottleneck for the deployment of large-scale AR image generators. The reported efficiency gains (5x memory, 6.6x throughput) are substantial and could significantly advance the practical usability of unified multimodal models on resource-constrained hardware.

**Weaknesses:**

- Generalizability: The paper's primary weakness is the limited scope of its validation. All analyses and experiments are conducted exclusively on the Janus-Pro model family. It remains unclear whether the core findings—the spatial-semantic dichotomy and margin column anchoring—are fundamental properties of visual AR generation or emergent properties specific to the Janus-Pro architecture. The claim needs to be validated on other visual AR models to be considered general.

- Static Head Classification: The classification of heads as spatial or semantic is performed offline and remains static throughout inference. This approach ignores the possibility that a head's function might be dynamic and context-dependent. A static policy may be suboptimal compared to a dynamic one.

- Hyperparameter Sensitivity: The method introduces several key hyperparameters, including the spatial window size $W$, the semantic budget $M$, and the classification threshold $\tau$. While the paper provides a good sensitivity analysis for $\tau$, it lacks a detailed discussion on the selection and sensitivity of $W$ and $M$. It is unclear how these are balanced to meet a specific cache budget and how performance is affected by their interplay.

**Questions:**

1. On Generalizability: Can you comment on the generalizability of your findings? Is there any evidence or strong reason to believe that the "spatial-semantic dichotomy" and "margin column anchoring" phenomena are also present in other visual AR models?

2. On Profiling Cost: Regarding the offline step, "Classify all heads via sparsity profiling": what is the computational cost of this process? Specifically, how many prompts were used to gather the statistics what hardware was used, and approximately how much time did this profiling take? Does this step pose a significant barrier to applying SSD to new models?

3. On Hyperparameters $W$ and $M$: Could you please elaborate on how the spatial window size $W$ and the semantic budget $M$ were selected? For instance, in the 20% cache budget scenario in Table 1, what were the typical values or ratio for $W$ and $M$? How sensitive is the model's performance to this allocation ratio?

4. On CFG Dependence: Your core analysis for identifying semantic injection (Figure 2(b)) is heavily dependent on CFG. Does the SSD framework remain effective, and does the margin anchoring phenomenon persist, during non-CFG sampling ($\gamma=1$) or at very low guidance scales?

5. On Static Classification: The head classification is static. Did you consider or experiment with a dynamic classification strategy, where a head's role could be re-evaluated or adapted based on the generation context?

---

> ### Author Response · Authors · 2025-11-23
> **Response to Reviewer ZLyX**
>
> Thanks for your thoughtful and positive review. We address the weaknesses and questions raised below.
>
> **Response to Weaknesses**
>
> ---
>
> - **[W1] Generalizability**
> - **[W1-Ans.]** We agree that validating beyond Janus-Pro is crucial. Our core findings—spatial-semantic dichotomy and margin column anchoring—appear fundamental to raster-scanned AR visual models, as they arise from inherent spatial structures and CFG mechanisms (Page 3). To demonstrate this, we extended analysis to LlamaGen-XL in the revised paper, confirming similar sparsity patterns and semantic concentration via attention visualizations and MSE plots (Appendix B, Page 13).
>
>     ---
>
> - **[W2] Static Head Classification**
> - **[W2-Ans]** We agree that head functions could theoretically be dynamic and context-dependent. However, our empirical analysis shows attention patterns *remain stable* across generation steps and prompts (Fig. 1). This consistency enables effective static offline classification without dynamic overhead, balancing simplicity and performance as evidenced by negligible quality loss in Table 1 (Page 7).
>
>     ---
>
> - **[W3] Hyperparameter Sensitivity**
> - **[W2-Ans.]** We acknowledge the role of hyperp. W (spatial window), M (semantic budget), and τ (threshold). These were selected based on the bimodal sparsity distribution (Fig. 2, Page 4), with τ empirically set to 0.8 to separate dense and sparse heads. We discuss in [Q3-Ans.].
>
> ---
>
> **Response to Questions**
>
> ---
>
> - **[Q1] On Generalizability**
> - **[Q1-Ans.]** As noted above, our findings generalize beyond Janus-Pro, as validated on LlamaGen-XL (Appendix B, Page 13). The phenomena stem from raster-order tokenization and CFG, common in AR models like LlamaGen.
>
>     ---
>
> - **[Q2] On Profiling Cost**
> - **[Q2-Ans.]** The offline head classification is computationally *lightweight.* We used 100 simple prompts (Page 2, randomly sampled from GenEval), inferencing them to collect average attention maps, and compute sparsity scores. This process runs quickly on modern hardware, ie., on an RTX 4090 GPU, it takes approximately 10 minutes. The low cost makes our framework applicable once per model without barriers.
>
>     ---
>
> - **[Q3] On Hyperparameters budget W and M**
> - **[Q3-Ans.]** For Table 1 comparisons (Page 7), we fixed W=32 for spatial heads across all methods. Under 20% budget, this yields ~0.2 cache ratio (e.g., recent=32, sink=90,  (32+90)/(32+576)≈0.2). Similar calculations apply to other budgets. Sensitivity analysis on head allocation ratios shows optimal performance at 0.6–0.8, with GenEval scores ranging from 0.65 (0.0 ratio) to 0.72 (0.8 ratio), confirming decoupling's efficacy.
>
>
>     | τ | 0.0 | 0.2 | 0.4 | 0.6 | **0.8** | 1.0 |
>     | --- | --- | --- | --- | --- | --- | --- |
>     | GenEval | 0.65 | 0.68 | 0.69 | 0.70 | **0.72** | 0.70 |
>
>     ---
>
> - **[Q4]On CFG Dependence**
> - **[Q4-Ans.]** SSD operates independently of CFG; Figur 2(b) (Page 4) is an interpretability tool visualizing semantic injection via MSE.
>
>     ---
>
> - **[Q5] On Static Classification:**
> - **[Q5-Ans.]** We considered dynamic strategies but deemed them unnecessary given fixed patterns (Figure 1, Page 2). Dynamics would add runtime costs without gains, as static SSD matches full-cache quality (Table 1, Pages 6-7).
>
> ---

---

> ### Author Response · Authors · 2025-11-23
> **Encouraging Discussion of Rebuttal**
>
> Dear Reviewer ZLyX:
>
> We hope that our detailed responses and the substantially revised manuscript address your concerns. If you find the improvements satisfactory, we would be grateful for a reconsideration of your evaluation. We greatly appreciate your constructive feedback and the time you’ve dedicated to reviewing our work.
>
> Best regards,
>
> Authors

---

### Meta-Review · Area_Chair_vdcU · 2026-01-12

**Summary:**

The reviewers' primary concerns center on the insufficient and questionable empirical validation of the proposed method. While the core idea of decoupling attention heads for KV cache compression is interesting and intuitive, the evidence supporting its effectiveness and generality is not yet at the ICLR standard.
The three main outstanding issues are:

1. Lack of Generalizability and Comprehensive Evaluation: All reviewers pointed out that the method was only fully evaluated on a single model family (Janus-Pro). Despite adding an analysis on a second model (LlamaGen-XL) during the rebuttal, the authors failed to provide a full evaluation of their method's performance on this model, a specific request from Reviewers hsdb and tLGE.

2. Missing Standard Evaluation Metrics: Reviewer hsdb strongly argued that for a lossy compression technique, standard distribution-based metrics like FID are crucial to assess image quality degradation. The authors initially dismissed this, and after agreeing to provide the results during the rebuttal period, they failed to do so.

3. Questionable Throughput Claims: Reviewer UEyc raised a critical and unanswered question about an anomaly in the baseline's reported throughput, which unexpectedly decreases as batch size increases. The authors' failure to address this point casts doubt on the validity of their experimental setup and the claimed 6.6x speedup, which is a core contribution.

Collectively, these unaddressed issues undermine the paper's claims of achieving significant efficiency gains with negligible performance loss in a generalizable manner.

**Reviewer Concerns:**

Addressed Concerns:

1. Novelty of Core Concepts: Reviewers hsdb and tLGE questioned the novelty, noting that spatial locality and attention sinks are known phenomena. The authors successfully clarified that their contribution is not in discovering these phenomena, but in the novel act of head decoupling: systematically identifying and applying distinct, tailored compression strategies to different head types within a single model. This reframing was accepted by the reviewers.

2. Discussion of Related Work: Reviewers UEyc and tLGE pointed out missing citations to parallel decoding methods (e.g., PAR, ZipAR). The authors addressed this by adding the citations and arguing that those methods are orthogonal (requiring training modifications or different decoding schemes), whereas their method is a training-free plug-in for standard AR decoding. This distinction was deemed reasonable.

Outstanding Concerns:
1. Generalizability: This was a unanimous concern (ZLyX, UEyc, hsdb, tLGE). While the authors added an analysis of attention patterns on LlamaGen-XL, this only partially addressed the issue. Reviewers hsdb and tLGE explicitly stated that this was insufficient and that a full application and performance evaluation on a second model family was necessary. The authors promised these results but did not deliver them, leaving the central claim of generalizability unsubstantiated.

2. Insufficient Evaluation Metrics: Reviewer hsdb’s request for FID/IS metrics was critical. The authors' failure to provide these results during the discussion period, after promising to do so, is a significant weakness. For a method that performs lossy compression, the lack of established distributional quality metrics makes it impossible to fully assess the "negligible performance loss" claim.

3. Anomalous Throughput Baseline: Reviewer UEyc’s follow-up question regarding the baseline model's throughput decreasing with increased batch size was a critical observation that questions the validity of the speedup experiments.

**Reviewer Scores:**

- Reviewer ZLyX (Initial Score: 6): This reviewer's primary concern was generalizability. While the authors' addition of LlamaGen analysis in the appendix does not seem to fully address. I think the scores will remain 6.

- Reviewer UEyc (Initial Score: 2): This reviewer would have certainly kept their score at 2 (Reject). Their pointed and critical question about the baseline's throughput anomaly was ignored, and the problem on generalizability is not solved. Therefore, I think the final score would be 2 or 4.

- Reviewer hsdb (Initial Score: 4): This reviewer would likely keep their score at 4. They explicitly stated that their concern about evaluation (W2) was not resolved and strongly recommended FID experiments. The authors' failure to deliver these results would confirm the paper's weakness in this crucial area.

- Reviewer tLGE (Initial Score: 4): This reviewer would also likely keep their score at 4. They acknowledged that their novelty concern was addressed but explicitly stated that the experimental evidence for generalizability was "still not fully convincing" and that a full application on LlamaGen was needed. The authors' failure to provide this would have confirmed their view that the paper is not ready for acceptance.

---

### Decision · Program_Chairs · 2026-01-26

Reject